# EMERGENCE OF FOVEAL IMAGE SAMPLING FROM LEARNING TO ATTEND IN VISUAL SCENES

**Brian Cheung, Eric Weiss, Bruno Olshausen**
Redwood Center
UC Berkeley
{bcheung,eaweiss,baolshausen}@berkeley.edu

## ABSTRACT

We describe a neural attention model with a learnable retinal sampling lattice. The model is trained on a visual search task requiring the classification of an object embedded in a visual scene amidst background distractors using the smallest number of fixations. We explore the tiling properties that emerge in the model's retinal sampling lattice after training. Specifically, we show that this lattice resembles the eccentricity dependent sampling lattice of the primate retina, with a high resolution region in the fovea surrounded by a low resolution periphery. Furthermore, we find conditions where these emergent properties are amplified or eliminated providing clues to their function.

## 1 INTRODUCTION

A striking design feature of the primate retina is the manner in which images are spatially sampled by retinal ganglion cells. Sample spacing and receptive fields are smallest in the fovea and then increase linearly with eccentricity, as shown in Figure 1. Thus, we have highest spatial resolution at the center of fixation and lowest resolution in the periphery, with a gradual fall-off in resolution as one proceeds from the center to periphery. The question we attempt to address here is, *why* is the retina designed in this manner - i.e., how is it beneficial to vision?

The commonly accepted explanation for this eccentricity dependent sampling is that it provides us with both high resolution and broad coverage of the visual field with a limited amount of neural resources. The human retina contains 1.5 million ganglion cells, whose axons form the sole output of the retina. These essentially constitute about 300,000 distinct samples of the image due to the multiplicity of cell types coding different aspects such as on vs. off channels (Van Essen & Anderson, 1995). If these were packed uniformly at highest resolution (120 samples/deg, the Nyquist-dictated sampling rate corresponding to the spatial-frequencies admitted by the lens), they would subtend an image area spanning just 5x5 deg$^2$. Thus we would have high-resolution but essentially tunnel vision. Alternatively if they were spread out uniformly over the entire monocular visual field spanning roughly 150 deg$^2$ we would have wide field of coverage but with very blurry vision, with each sample subtending 0.25 deg (which would make even the largest letters on a Snellen eye chart illegible). Thus, the primate solution makes intuitive sense as a way to achieve the best of both of these worlds. However we are still lacking a quantitative demonstration that such a sampling strategy emerges as the optimal design for subserving some set of visual tasks.

Here, we explore what is the optimal retinal sampling lattice for an (overt) attentional system performing a simple visual search task requiring the classification of an object. We propose a learnable retinal sampling lattice to explore what properties are best suited for this task. While evolutionary pressure has tuned the retinal configurations found in the primate retina, we instead utilize gradient descent optimization for our in-silico model by constructing a fully differentiable dynamically controlled model of attention.

Our choice of visual search task follows a paradigm widely used in the study of overt attention in humans and other primates (Geisler & Cormack, 2011). In many forms of this task, a single target is randomly located on a display among distractor objects. The goal of the subject is to find the target as rapidly as possible. Itti & Koch (2000) propose a selection mechanism based on manually

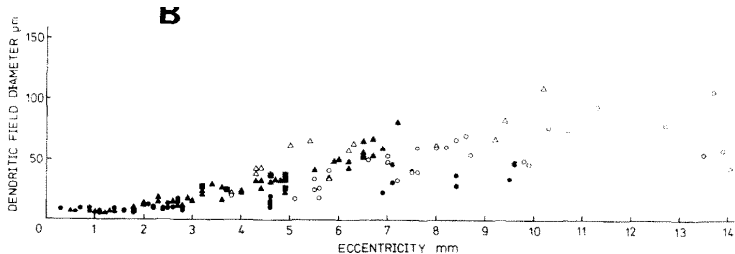

Figure 1: Receptive field size (dendritic field diameter) as a function of eccentricity of Retinal Ganglion Cells from a macaque monkey (taken from Perry et al. (1984)).

defined low level features of real images to locate various search targets. Here the neural network must learn what features are most informative for directing attention.

While neural attention models have been applied successfully to a variety of engineering applications (Bahdanau et al., 2014; Jaderberg et al., 2015; Xu et al., 2015; Graves et al., 2014), there has been little work in relating the properties of these attention mechanisms back to biological vision. An important property which distinguishes neural networks from most other neurobiological models is their ability to learn *internal* (latent) features directly from data.

But existing neural network models specify the input sampling lattice *a priori*. Larochelle & Hinton (2010) employ an eccentricity dependent sampling lattice mimicking the primate retina, and Mnih et al. (2014) utilize a multi scale glimpse window' that forms a piece-wise approximation of this scheme. While it seems reasonable to think that these design choices contribute to the good performance of these systems, it remains to be seen if this arrangement emerges as the optimal solution.

We further extend the learning paradigm of neural networks to the *structural* features of the glimpse mechanism of an attention model. To explore emergent properties of our learned retinal configurations, we train on artificial datasets where the factors of variation are easily controllable. Despite this departure from biology and natural stimuli, we find our model learns to create an eccentricity dependent layout where a distinct central region of high acuity emerges surrounded by a low acuity periphery. We show that the properties of this layout are highly dependent on the variations present in the task constraints. When we depart from physiology by augmenting our attention model with the ability to spatially rescale or zoom on its input, we find our model learns a more uniform layout which has properties more similar to the glimpse window proposed in Jaderberg et al. (2015); Gregor et al. (2015). These findings help us to understand the task conditions and constraints in which an eccentricity dependent sampling lattice emerges.

## 2 RETINAL TILING IN NEURAL NETWORKS WITH ATTENTION

Attention in neural networks may be formulated in terms of a differentiable feedforward function. This allows the parameters of these models to be trained jointly with backpropagation. Most formulations of visual attention over the input image assume some structure in the kernel filters. For example, the recent attention models proposed by Jaderberg et al. (2015); Mnih et al. (2014); Gregor et al. (2015); Ba et al. (2014) assume each kernel filter lies on a rectangular grid. To create a learnable retinal sampling lattice, we relax this assumption by allowing the kernels to tile the image independently.

### 2.1 GENERATING A GLIMPSE

We interpret a glimpse as a form of routing where a subset of the visual scene $U$ is sampled to form a smaller output glimpse $G$. The routing is defined by a set of kernels $k[\bullet](s)$, where each kernel $i$ specifies which part of the input $U[\bullet]$ will contribute to a particular output $G[i]$. A control variable $s$

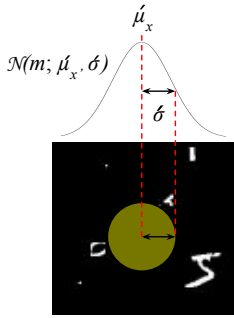

Figure 2: Diagram of single kernel filter parameterized by a mean $\acute{\mu}$ and variance $\acute{\sigma}$.

is used to control the routing by adjusting the position and scale of the entire array of kernels. With this in mind, many attention models can be reformulated into a generic equation written as

$$G[i] = \sum_n^H \sum_m^W U[n, m]k[m, n, i](s) \tag{1}$$

where $m$ and $n$ index input pixels of $U$ and $i$ indexes output glimpse features. The pixels in the input image $U$ are thus mapped to a smaller glimpse $G$.

## 2.2 RETINAL GLIMPSE

The centers of each kernel filter $\acute{\mu}[i]$ are calculated with respect to control variables $s_c$ and $s_z$ and learnable offset $\mu[i]$. The control variables specify the position and zoom of the entire glimpse. $\mu[i]$ and $\sigma[i]$ specify the position and spread respectively of an individual kernel $k[-, -, i]$. These parameters are learned during training with backpropagation. We describe how the control variables are computed in the next section. The kernels are thus specified as follows:

$$\acute{\mu}[i] = (s_c - \mu[i])s_z \tag{2}$$
$$\acute{\sigma}[i] = \sigma[i]s_z \tag{3}$$
$$k[m, n, i](s) = \mathcal{N}(m; \acute{\mu}_x[i], \acute{\sigma}[i])\mathcal{N}(n; \acute{\mu}_y[i], \acute{\sigma}[i]) \tag{4}$$

We assume kernel filters factorize between the horizontal $m$ and vertical $n$ dimensions of the input image. This factorization is shown in equation 4, where the kernel is defined as an isotropic gaussian $\mathcal{N}$. For each kernel filter, given a center $\acute{\mu}[i]$ and scalar variance $\acute{\sigma}[i]$, a two dimensional gaussian is defined over the input image as shown in Figure 2. These gaussian kernel filters can be thought of as a simplified approximation to the receptive fields of retinal ganglion cells in primates (Van Essen & Anderson, 1995).

While this factored formulation reduces the space of possible transformations from input to output, it can still form many different mappings from an input $U$ to output $G$. Figure 3B shows the possible windows which an input image can be mapped to an output $G$. The yellow circles denote the central location of a particular kernel while the size denotes the standard deviation. Each kernel maps to one of the outputs $G[i]$.

Positional control $s_c$ can be considered analogous to the motor control signals which executes saccades of the eye, whereas $s_z$ would correspond to controlling a zoom lens in the eye (which has no counterpart in biology). In contrast, *training* defines *structural* adjustments to individual kernels which include its position in the lattice as well as its variance. These adjustments are only possible during training and are fixed afterwards.Training adjustments can be considered analagous to the incremental adjustments in the layout of the retinal sampling lattice which occur over many generations, directed by evolutionary pressure in biology.

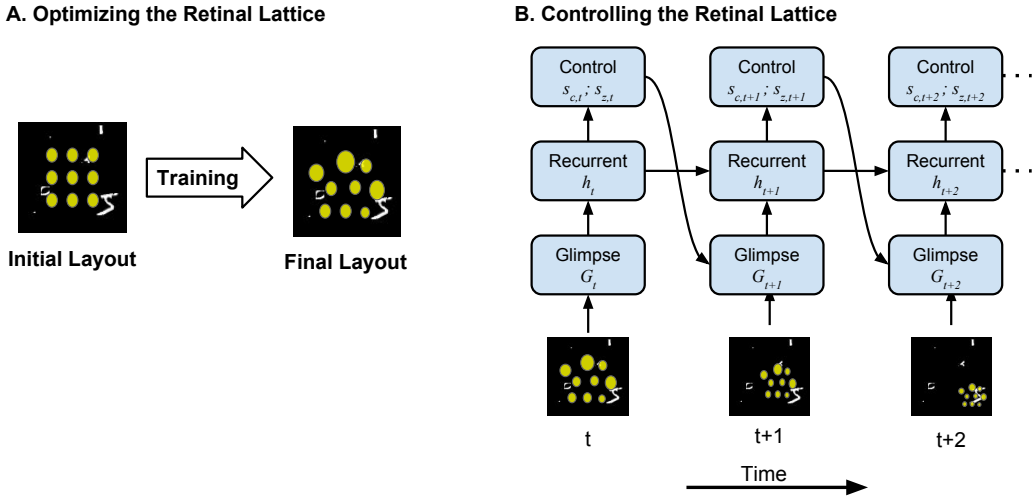

Figure 3: **A**: Starting from an initial lattice configuration of a uniform grid of kernels, we learn an optmized configuration from data. **B**: Attentional fixations generated during inference in the model, shown unrolled in time (after training).

Table 1: Variants of the neural attention model

| Ability | Fixed Lattice | Translation Only | Translation and Zoom |
|---|---|---|---|
| Translate retina via $s_{c,t}$ | ✓ | ✓ | ✓ |
| Learnable $\mu[i]$, $\sigma[i]$ | | ✓ | ✓ |
| Zoom retina via $s_{z,t}$ | | | ✓ |

## 3 RECURRENT NEURAL ARCHITECTURE FOR ATTENTION

A glimpse at a specific timepoint, $G_t$, is processed by a fully-connected recurrent network $f_{rnn}()$.

$$h_t = f_{rnn}(G_t, h_{t-1}) \tag{5}$$
$$[s_{c,t}; s_{z,t}] = f_{control}(h_t) \tag{6}$$

The global center $s_{c,t}$ and zoom $s_{z,t}$ are predicted by the control network $f_{control}()$ which is parameterized by a fully-connected neural network.

In this work, we investigate three variants of the proposed recurrent model:

- **Fixed Lattice:** The kernel parameters $\mu[i]$ and $\sigma[i]$ for each retinal cell are *not* learnable. The model can only translate the kernel filters $s_{c,t} = f_{control}(h_t)$ and the global zoom is fixed $s_{z,t} = 1$.

- **Translation Only:** Unlike the fixed lattice model, $\mu[i]$ and $\sigma[i]$ are learnable (via backpropagation).

- **Translation and Zoom:** This model follows equation 6 where it can both zoom and translate the kernels.

A summary for comparing these variants is shown in Table 1.

Prior to training, the kernel filters are initialized as a 12x12 grid (144 kernel filters), tiling uniformly over the central region of the input image and creating a retinal sampling lattice as shown in Figure 5 before training. Our recurrent network, $f_{rnn}$ is a two layer traditional recurrent network with 512-512 units. Our control network, $f_{control}$ is a fully-connected network with 512-3 units (x,y,zoom)

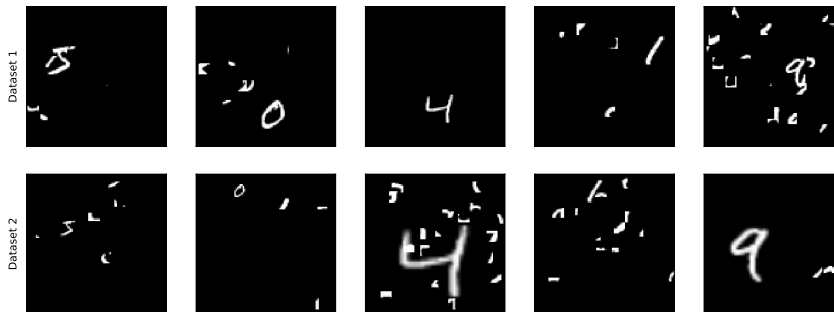

Figure 4: *Top Row*: Examples from our variant of the cluttered MNIST dataset (a.k.a Dataset 1). *Bottom Row*: Examples from our dataset with variable sized MNIST digits (a.k.a Dataset 2).

in each layer. Similarly, our prediction networks are fully-connected networks with 512-10 units for predicting the class. We use ReLU non-linearities for all hidden unit layers.

Our model as shown in Figure 3C are differentiable and trained end-to-end via backpropagation through time. Note that this allows us to train the control network indirectly from signals backpropagated from the task cost. For stochastic gradient descent optimization we use Adam (Kingma & Ba, 2014) and construct our models in Theano (Bastien et al., 2012).

## 4 DATASETS AND TASKS

### 4.1 MODIFIED CLUTTERED MNIST DATASET

Example images from of our dataset are shown in Figure 4. Handwritten digits from the original MNIST dataset LeCun & Cortes (1998) are randomly placed over a 100x100 image with varying amounts of distractors (clutter). Distractors are generated by extracting random segments of non-target MNIST digits which are placed randomly with uniform probability over the image. In contrast to the cluttered MNIST dataset proposed in Mnih et al. (2014), the number of distractors for each image varies randomly from 0 to 20 pieces. This prevents the attention model from learning a solution which depends on the number 'on' pixels in a given region. In addition, we create another dataset (Dataset 2) with an additional factor of variation: the original MNIST digit is randomly resized by a factor of 0.33x to 3.0x. Examples of this dataset are shown in the second row of Figure 4.

### 4.2 VISUAL SEARCH TASK

We define our visual search task as a recognition task in a cluttered scene. The recurrent attention model we propose must output the class $\hat{c}$ of the single MNIST digit appearing in the image via the prediction network $f_{predict}()$. The task loss, $\mathcal{L}$, is specified in equation 8. To minimize the classification error, we use cross-entropy cost:

$$\hat{c}_{t,n} = f_{predict}(h_{t,n}) \tag{7}$$

$$\mathcal{L} = \sum_n^N \sum_t^T c_n log(\hat{c}_{t,n}) \tag{8}$$

Analolgous to the visual search experiments performed in physiological studies, we pressure our attention model to accomplish the visual search as quickly as possible. By applying the task loss to every timepoint, the model is forced to accurately recognize and localize the target MNIST digit in as few iterations as possible. In our classification experiments, the model is given $T = 4$ glimpses.

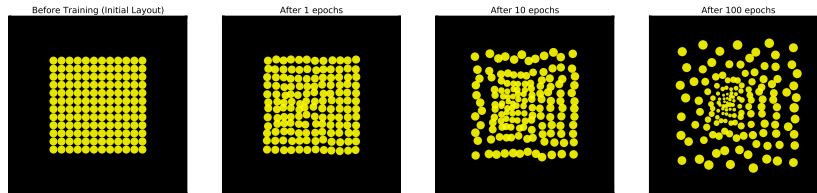

Figure 5: The sampling lattice shown at four different stages during training for a Translation Only model, from the initial condition (left) to final solution (right). The radius of each dot corresponds to the standard deviation $\sigma_i$ of the kernel.

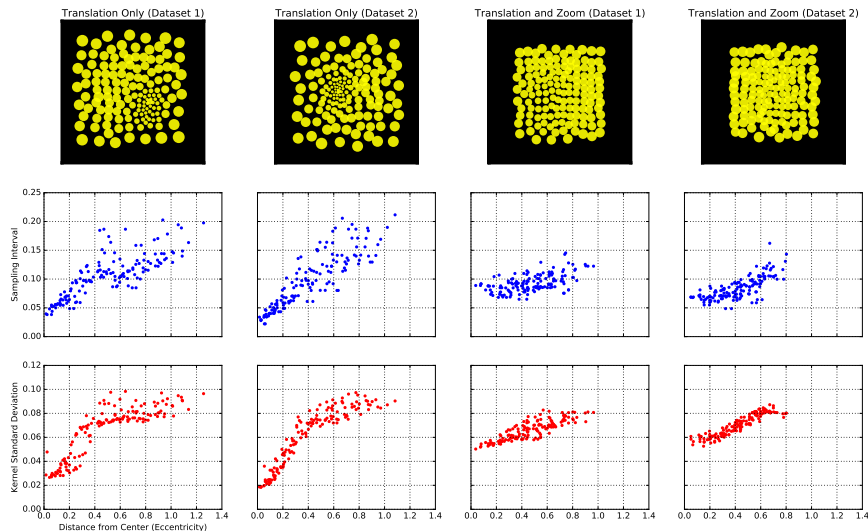

Figure 6: *Top*: Learned sampling lattices for four different model configurations. *Middle*: Resolution (sampling interval) and *Bottom*: kernel standard deviation as a function of eccentricity for each model configuration.

## 5    RESULTS

Figure 5shows the layouts of the learned kernels for a Translation Only model at different stages during training. The filters are smoothly transforming from a uniform grid of kernels to an eccentricity dependent lattice. Furthermore, the kernel filters spread their individual centers to create a sampling lattice which covers the full image. This is sensible as the target MNIST digit can appear anywhere in the image with uniform probability.

When we include variable sized digits as an additional factor in the dataset, the translation only model shows an even greater diversity of variances for the kernel filters. This is shown visually in the first row of Figure 6. Furthermore, the second row shows a highly dependent relationship between the sampling interval and standard deviatoin of the retinal sampling lattice and eccentricity from the center. This dependency increases when training on variable sized MNIST digits (Dataset 2). This

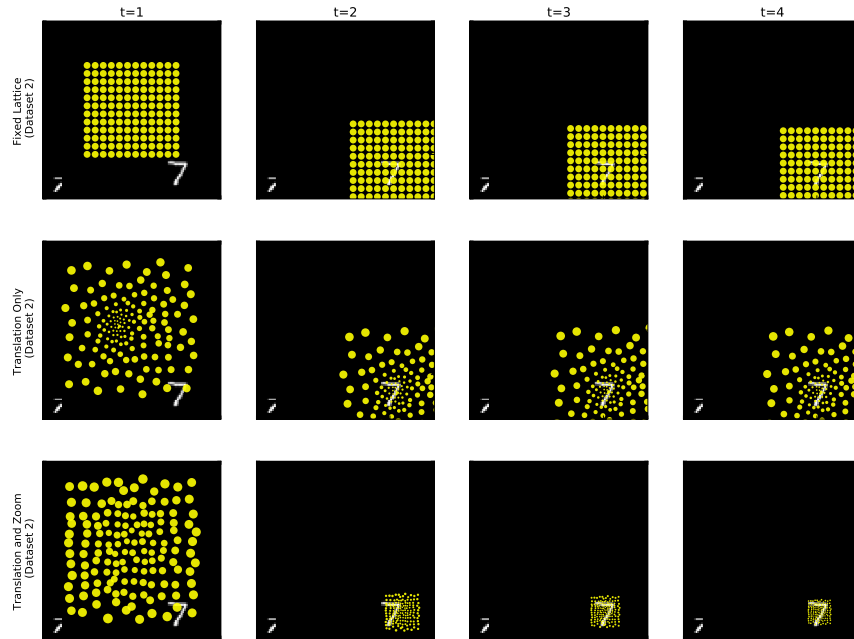

Figure 7: Temporal rollouts of the retinal sampling lattice attending over a test image from Cluttered MNIST (Dataset 2) after training.

relationship has also been observed in the primate visual system (Perry et al., 1984; Van Essen & Anderson, 1995).

When the proposed attention model is able to zoom its retinal sampling lattice, a very different layout emerges. There is much less diversity in the distribution of kernel filter variances as evidenced in Figure 6. Both the sampling interval and standard deviation of the retinal sampling lattice have far less of a dependence on eccentricity. As shown in the last column of Figure 6, we also trained this model on variable sized digits and noticed no significant differences in sampling lattice configuration.

Figure 7 shows how each model variant makes use of its retinal sampling lattice after training. The strategy each variant adopts to solve the visual search task helps explain the drastic difference in lattice configuration. The translation only variant simply translates its high acuity region to recognize and localize the target digit. The translation and zoom model both rescales and translates its sampling lattice to fit the target digit. Remarkably, Figure 7 shows that both models detect the digit early on and make minor corrective adjustments in the following iterations.

Table 2 compares the classification performance of each model variant on the cluttered MNIST dataset with fixed sized digits (Dataset 1). There is a significant drop in performance when the retinal sampling lattice is fixed and not learnable, confirming that the model is benefitting from learning the high-acuity region. The classification performance between the Translation Only and Translation and Zoom model is competitive. This supports the hypothesis that the functionality of a high acuity region with a low resolution periphery is similar to that of zoom.

Table 2: Classification Error on Cluttered MNIST

| Sampling Lattice Model | Dataset 1 (%) | Dataset 2 (%) |
|---|---|---|
| Fixed Lattice | 11.8 | 31.9 |
| Translation Only | 5.1 | 24.4 |
| Translation and Zoom | 4.0 | 24.1 |

## 6 CONCLUSION

When constrained to a glimpse window that can translate only, similar to the eye, the kernels converge to a sampling lattice similar to that found in the primate retina (Curcio & Allen, 1990; Van Essen & Anderson, 1995). This layout is composed of a high acuity region at the center surrounded by a wider region of low acuity. Van Essen & Anderson (1995) postulate that the linear relationship between eccentricity and sampling interval leads to a form of scale invariance in the primate retina. Our results from the Translation Only model with variable sized digits supports this conclusion. Additionally, we observe that zoom appears to supplant the need to learn a high acuity region for the visual search task. This implies that the high acuity region serves a purpose resembling that of a zoomable sampling lattice. The low acuity periphery is used to detect the search target and the high acuity 'fovea' more finely recognizes and localizes the target. These results, while obtained on an admittedly simplified domain of visual scenes, point to the possibility of using deep learning as a tool to explore the optimal sample tiling for a retinal in a data driven and task-dependent manner. Exploring how or if these results change for more challenging tasks in naturalistic visual scenes is a future goal of our research.

ACKNOWLEDGMENTS

We would like to acknowledge everyone at the Redwood Center for their helpful discussion and comments. We gratefully acknowledge the support of NVIDIA Corporation with the donation of the Tesla K40 GPUs used for this research.

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
