# Peer review of "Emergence of foveal image sampling from learning to attend in visual scenes"

_ICLR 2017 — accepted_

[Official Review · AnonReviewer3 · rating 6 · confidence 5 · 16 Dec 2016]
**A good apporach to attention-based models**

This paper proposed a neural attention model which has a learnable and differentiable sampling lattice. The work is well motivated as few previous work focus on learning the sampling lattice but with a fixed lattice. This work is quite similar to Spatial Transformer Networks (Jaderberg 2015), but the sampling lattice is learned by the model. The experiments showed that the model can learn a meaning lattice to the visual search task where the sampling lattice looks similar to human being's. 

The main concern of the paper is that experiments are not sufficient. The paper only reports the results on a modified clustered MNIST dataset. It would be more interesting if the authors could conduct  the model on real datasets, such as Toronto Face dataset, CUB bird dataset and SVHN. For example, for the Face dataset, it would be nice if the model can learn to attend different parts of the face for expression recognition, or attend different part of birds for fine-grained classification. Since the authors replied in the pre-review question that the model can learn meaningful lattice on MSCOCO dataset, I think it would be better to add that results into the paper.

Another drawback of the model is that the paper only compare with different variants of itselves. I suggest that this paper should compare with  Spatial Transformer Networks, DRAW, etc., on the same dataset to show the advantage of the learned sampling lattice.

[Official Review · AnonReviewer2 · rating 5 · confidence 4 · 20 Dec 2016]

The paper presented an extension to the current visual attention model that learns a deformable sampling lattice.  Comparing to the fixed sampling lattice from previous works, the proposed method shows different sampling strategy can emerge depending on the visual classification tasks. The authors empirically demonstrated the learnt sampling lattice outperforms the fixed strategies. More interestingly, when the attention mechanism is constrained  to be translation only, the proposed model learns a sampling lattice resembles the retina found in the primate retina.  


Pros:
+ The paper is generally well organized and written 
+ The qualitative analysis in the experimental section is very comprehensive.

Cons:
-  The paper could benefit substantially from additional experiments on different datasets.
-  It is not clear from the tables the proposed learnt sampling  lattice offer any computation benefit when comparing to  a fixed sampling strategy with zooming capability, e.g. the one used in DRAW model.

Overall, I really like the paper. I think the experimental section can be improved by additional experiments and more quantitative analysis with other baselines. Because the current revision of the paper only shows experiments on digit dataset with black background, it is hard to generalize the finding or even to verify the claims in the paper, e.g.  linear relationship
between eccentricity and sampling interval leads to the primate retina, from the results on a single dataset.

[Official Review · AnonReviewer1 · rating 6 · confidence 4 · 24 Dec 2016]
**Solid hypothesis but experiments leave doubts**

This paper presents a succinct argument that the principle of optimizing receptive field location and size in a simulated eye that can make saccades with respect to a classification error of images of data whose labels depend on variable-size and variable-location subimages, explains the existence of a foveal area in e.g. the primate retina.

The argument could be improved by using more-realistic image data and drawing more direct correspondence with the number, receptive field sizes and eccentricities of retinal cells in e.g. the macaque, but the authors would then face the challenge of identifying a loss function that is both biologically plausible and supportive of their claim.

The argument could also be improved by commenting on the timescales involved. Presumably the density of the foveal center depends on the number of of saccades allowed by the inference process, as well as the size of the target sub-images, and also has an impact on the overall classification accuracy.

Why does the classification error rate of dataset 2 remain stubbornly at 24%? This seems so high that the model may not be working the way we’d like it to. It seems that the overall argument of the paper pre-supposes that the model can be trained to be a good classifier. If there are other training strategies or other models that work better and differently, then it raises the question of why do our eyes and visual cortex not work more like *those ones* if evolutionary pressures are applying the same pressure as our training objective.

Why does the model with zooming powers out-do the translation-only model on dataset 1 (where all target images are the same size) and tie the translation-only model dataset 2 (where the target images have different sizes, for which the zooming model should be tailor-made?). Between this strange tie and the high classification rate on Dataset 2, I wonder if maybe one or both models isn’t being trained to its potential, which would undermine the overall claim.

Comparing this model to other attention models (e.g. spatial transformer networks, DRAW) would be irrelevant to what I take to be the main point of the paper, but it would address the potential concerns above that training just didn’t go very well, or there was some problem with the model parameterization that could be easily fixed.

[Final Decision · Program Chairs · 06 Feb 2017]
**ICLR committee final decision**

This was a borderline case. All reviewers and the AC appeared to find the paper interesting, while having some reservations. Given the originality of the work, the PCs decided to lean toward acceptance. We do encourage however the authors to revise their paper based on reviewer feedback as much as possible, to increase its potential for impact.